# Exploring Cell Migration Mechanisms in Cancer: From Wound Healing Assays to Cellular Automata Models

**DOI:** 10.3390/cancers15215284

**Published:** 2023-11-03

**Authors:** Giorgia Migliaccio, Rosalia Ferraro, Zhihui Wang, Vittorio Cristini, Prashant Dogra, Sergio Caserta

**Affiliations:** 1Dipartimento di Ingegneria Chimica, dei Materiali e Della Produzione Industriale, Università Degli Studi di Napoli Federico II, 80125 Naples, Italy; giorgi.migliaccio@studenti.unina.it (G.M.); rosalia.ferraro@unina.it (R.F.); 2CEINGE Biotecnologie Avanzate, Via Gaetano Salvatore, 80145 Naples, Italy; 3Mathematics in Medicine Program, Department of Medicine, Houston Methodist Research Institute, Houston, TX 77030, USA; zwang@houstonmethodist.org (Z.W.); vcristini@houstonmethodist.org (V.C.); pdogra@houstonmethodist.org (P.D.); 4Neal Cancer Center, Houston Methodist Research Institute, Houston, TX 77030, USA; 5Department of Imaging Physics, University of Texas MD Anderson Cancer Center, Houston, TX 77030, USA; 6Department of Physiology and Biophysics, Weill Cornell Medical College, New York, NY 10065, USA; 7Physiology, Biophysics, and Systems Biology Program, Graduate School of Medical Sciences, Weill Cornell Medicine, New York, NY 10065, USA

**Keywords:** wound healing, cancer invasion, cell migration, cellular automata model, digital twin

## Abstract

**Simple Summary:**

Cell migration is a key factor in the spread of metastatic tumors and a major contributor to cancer-related mortality. However, our comprehension of the underlying mechanisms remains incomplete. In this study, we utilized a wound healing assay to explore the migration and invasion of cancer cells in the context of metastasis. We developed a computational model using cellular automata, rigorously calibrated and validated with in vitro data from both tumor and non-tumor cell lines, offering a potent resource. This novel approach is of immense value to the pharmaceutical sector for discovering compounds that can impede cell migration, evaluating the efficacy of potential drugs to hinder cancer invasion, and assessing immune system responses. It stands as a breakthrough in the quest for more effective cancer therapies.

**Abstract:**

Purpose: Cell migration is a critical driver of metastatic tumor spread, contributing significantly to cancer-related mortality. Yet, our understanding of the underlying mechanisms remains incomplete. Methods: In this study, a wound healing assay was employed to investigate cancer cell migratory behavior, with the aim of utilizing migration as a biomarker for invasiveness. To gain a comprehensive understanding of this complex system, we developed a computational model based on cellular automata (CA) and rigorously calibrated and validated it using in vitro data, including both tumoral and non-tumoral cell lines. Harnessing this CA-based framework, extensive numerical experiments were conducted and supported by local and global sensitivity analyses in order to identify the key biological parameters governing this process. Results: Our analyses led to the formulation of a power law equation derived from just a few input parameters that accurately describes the governing mechanism of wound healing. This groundbreaking research provides a powerful tool for the pharmaceutical industry. In fact, this approach proves invaluable for the discovery of novel compounds aimed at disrupting cell migration, assessing the efficacy of prospective drugs designed to impede cancer invasion, and evaluating the immune system’s responses.

## 1. Introduction

Metastasis, which accounts for approximately 67% of cancer-related fatalities, remains an enigmatic and complex process involving the dissemination of cancer cells from their primary tumor site [1]. Despite its pivotal role in the formation of secondary malignant growths, a comprehensive understanding of metastasis continues to elude the scientific community. Therefore, unraveling the intricacies of cell migration takes on paramount significance as it forms the foundation for comprehending and, potentially, controlling metastasis [2,3].

An array of methods has been employed to study cell migration [4]: the in vitro wound healing (WH) assay [5,6,7], often referred to as the in vitro scratch assay; Boyden chamber assays [8,9]; live-cell imaging for real-time observation [10,11,12]; single-cell tracking for quantitative analysis [13,14]; gene expression profiling to identify key regulators [15,16]; and protein–protein interaction analysis to examine the molecular interactions involved in cell migration [17]. The WH assay method, characterized by its popularity, cost-effectiveness, and standardized in vitro approach, involves inducing a controlled scratch within a confluent cell monolayer, typically through mechanical, thermal, or chemical damage [18]. Following injury, cells initiate migration onto the wound bed, thereby contributing to the restoration of the epidermal barrier’s structure and function [19]. This overall process incorporates cell migration, proliferation, and differentiation. In some instances, to mitigate the potential confounding effects of cell proliferation, a low dose of the proliferation inhibitor mitomycin C is employed [20,21,22]. In addition, other common strategies include reducing the percentage of FBS in the media in order to decrease the proliferation rate or employing shorter timepoints (e.g., 8–12 h) for measuring wound size after the initial wound creation. This approach minimizes the impact of cell proliferation on wound closure.

As an alternative to traditional in vitro approaches, computational modeling emerges as an efficient tool for the quantitative exploration of biological systems [23,24,25,26] and the management of extensive experimental datasets. A particularly renowned model employed in the context of the WH phenomenon is the Fisher–Kolmogorov (FK) equation [27]. This mathematical framework delineates the evolution of cell density (u (cells/μm²)) in both the spatial (x (μm)) and temporal (t (h)) dimensions, amalgamating cell migration and proliferation processes, which are characterized as Fickian diffusion and logistic growth, respectively, such that $\frac{\partial u}{\partial t}=D\frac{\partial^{2}u}{\partial x^{2}}+\mathrm{ku}\left(1-\frac{u}{\hat{u}}\right)$ [28]. The FK model serves as a foundational platform, facilitating the development of various adaptations aimed at simulating complex biological phenomena [28,29,30,31]. These adaptations incorporate integrodifferential or Navier–Stokes equations and fluid dynamics models to describe both macroscopic [32] and microscopic aspects of biological systems [33,34,35].

Since the 1970s, an innovative approach rooted in the development of discrete mechanistic models, known as cellular automata (CA), has gained prominence. CA, celebrated for their simplicity, find application across diverse domains, spanning from urban system evolution [36,37] to material erosion and various biological processes [38,39,40], including the dynamics of the epithelial system [41], wound repair [42], viral infections [43], tumor response to therapies [44,45], and tumor metabolism [46]. CA models are characterized as spatially and temporally discrete systems governed by a set of rules rooted in fundamental biophysical phenomena, dictating the behavior of individual cells and their interactions within a defined spatial neighborhood [47]. This cellular-based modeling approach empowers the description of complex systems through simple relationships among constituent components, distinguishing itself from continuum models.

In this study, we introduce a CA model developed as a Digital Twin (DT) of the in vitro WH assay, enabling precise predictions of migration rates. The model’s validation was achieved through direct comparisons with in vitro data collected from previous experimental studies based on four different cell lines. Subsequently, the model underwent validation using five additional cell lines, four of which were obtained from the previously published literature, while the fifth was obtained from a novel, unpublished, experimental campaign. The overall number of nine different cell lines was intentionally chosen to ensure extended validation of the CA here developed across a wide spectrum of cell types and tissues. Notably, this validation encompassed various cell types, including epithelial and fibroblast cells, sourced from both murine and human origins, and represented both tumoral and non-tumoral cells. The results demonstrated good agreement between the model predictions and the experimental data.

Combining principles from the realms of biology and physics with modeling finds widespread utility in the pharmaceutical industry for identifying novel compounds targeting cell migration and evaluating the effectiveness of potential drugs designed to inhibit cancer invasion and evaluate the immune system’s response, thereby enhancing our understanding of potential clinical treatments [48].

## 2. Materials and Methods

In the upcoming section, we will provide a comprehensive breakdown of the structure of the CA model employed for simulating in vitro wound healing. To enhance clarity, the model description is organized into several subsections. Initially, we will introduce the model domain and outline the rules governing cellular dynamics. To facilitate a thorough comprehension of the model, a separate subsection will be exclusively devoted to statistical and sensitivity analyses.

### 2.1. In Vitro Experiments

The CA model was validated by direct comparison of in silico predictions with in vitro experimental data [5,31,49,50,51,52]. Four different cell lines were considered from previous works by our group: HT-1080 human fibrosarcoma cells [5], MDA-MB-231 [31,51] and MDA-MB-468 [51] breast cancer cell lines, and HaCaT [49,50] human keratinocyte cell line.

In our in vitro experiments, the quantification of WH dynamics was carried out by tracking the changes in the cell-free region’s area over time (A), which was then normalized to its initial value at time 0 (A_0_). To accomplish this, we employed custom-made automated image analysis software that relied on image variance analysis to identify the wound’s edge as the boundary of the cell-free area. Wound closure rate (α), obtained as the slope of the linear reduction in A/A_0_ as a function of time, was used as the key parameter for the comparison between in silico and in vitro experiments. The same approach was used to compare our data to previous in vitro experiments collected from the literature. For further elaboration on this technique, please refer to our previous publication [51].

### 2.2. Model Development

In the following section, we will provide a detailed presentation of the CA model used for simulating in vitro healing.

#### 2.2.1. Domain Building

The cellular monolayer of the WH assay was represented in silico by a two-dimensional lattice of square elements (N × N), where each element (with coordinates i,j) had a value of 1 or 0, depending on whether it was empty or occupied by a cell.

Thus, the dimensions of each element of the lattice correspond to the average size of a single cell (δ) (Figure 1a). As shown in Figure 1a, the initial configuration (at time t = 0) was characterized by two lateral cell domains simulating the edge of the wound (referred to as left (L) and right (R)) and one central domain mimicking the cell-free region (referred to as wound (W)). The dimension of W along the x-direction was defined by the mean value of the length of the in vitro wound (b_0_) divided by δ. Similarly, the initial number of cells in L and R, randomly distributed, was calculated from the cell density (ρ) in the in vitro experiments by assuming the L and R lengths as half of the dimension of W. As the simulation time went on, cells moved toward the wound from the edges, where a periodic boundary condition was set to guarantee constant cell density. The periodic boundary simulated an infinite space for the surrounding cells; in the script, the first and last columns of the lattice were repopulated with cells at each time step in case any vacancy was generated during the simulation by cell migration steps.

#### 2.2.2. Rules Governing Cellular Dynamics

CA evolved over time and space based on a set of simple rules associated with the two primary processes of migration and proliferation, which govern the dynamic evolution of living tissues. These processes will be discussed in detail in the following sections.

Migration and proliferation

When confronted with available empty spaces within the lattice, cells possess the flexibility to opt for either migration or proliferation, effectively occupying a chosen site among the nearest vacant lattice locations, as shown in Figure 1b,c. Within the CA lattice, the determination between migration and proliferation hinges on the estimated probabilities assigned to each process. In detail, migration probability is defined as $P_{m}=\frac{1/T_{m}}{1/T_{m}+1/T_{d}}$, where T_m_ is the time necessary for a cell to move a distance equal to its characteristic size δ. T_m_ can be calculated using the random motility coefficient as $T_{m}=\delta^{2}/D$, where D is the constant diffusion rate of cells. Similarly, proliferation probability is given by $P_{d}=\frac{1/T_{d}}{1/T_{d}+1/T_{m}}$, where T_d_ is the doubling time of a given cell line, typically available in the literature. Since, in a single step, each cell may either migrate or proliferate, the sum of these two probabilities is 1.

To determine the action for each cell in every time step, a stochastic process is employed. For each cell, a random value, denoted as “λ”, is generated from a uniform probability distribution within the range [0, 1]. Subsequently, this random value “λ” is compared to a P_m_. If λ < $P_{m}$ and a neighboring location is available, the cell will migrate; otherwise, the cell will proliferate.

This stochastic decision-making process, based on probabilistic comparisons, governs the behavior of each cell in the model, dynamically determining whether it should engage in migration or proliferation at each time step.

2.Quiescence

In cases where there are no available empty neighboring locations for cells to move into, the cells enter a quiescent state, unable to execute either migration or proliferation. This quiescent state is implemented to replicate the biological phenomenon of contact inhibition [53], where cells cease their movement and division when they are in close contact with neighboring cells.

The flow chart of the algorithm is reported in Figure 1d and described below:

(1)Spatial domain discretization and initialization of cell positions.(2)Testing for empty neighbors for every occupied element.(3)Random number (λ) assignment to the occupied CA elements to decide the actions of cells:
If λ > $P_{d}$, the actual site of the cell of interest will remain occupied by the cell, and a daughter cell will be placed in a randomly chosen empty site among the neighbors.If λ < $P_{m}$, the site of the cell of interest will become empty, and a neighboring empty site will become occupied.(4)Lattice updates according to the selected actions based on probabilities.(5)Stop if the wound was healed; otherwise, proceed to next time step and return to (1).

In wound repair, the direction of migration is affected by the presence of the damage due to the lack of contact inhibition [54] and chemical stimuli (i.e., nutrients, catabolites, etc.) [13]. To reproduce a chemical stimulus from the opposite edge of the wound, pushing the cells to move in the direction of the wound [13], the number of sites available in the opposite direction was reduced (from 9 to 7). However, the approach developed in our CA can be flexibly used to also describe different possible external stimuli, such as chemotaxis; for example, the presence of an external chemoattractant effect was analyzed in the Appendix A. In our work, an isotropic condition was assumed.

### 2.3. Parameter Sensitivity Analysis

To understand the effect of the biological and physical characteristics of the system on wound healing, parameter sensitivity analysis was performed. The model parameters of interest (T_m_, T_d_, ρ, δ, b_0_) were evaluated for their effect on the kinetics of wound closure (v, α, D, and characteristic times of the assay). The characteristic times of wound closure, typically measured experimentally in the literature, were T_half_ and T_closure_, defined as the times required for a closure of 50% and 100% of the original wound area A_0_, respectively.

In this analysis, the model parameters were subjected to perturbations within predefined ranges around their baseline values, and the resulting impact of these perturbations on the output variables was quantified. The specific range of physiologic variation for each of the parameters mentioned above, obtained from the literature, along with their definitions, is reported in Table 1.

#### Global Sensitivity Analysis

A global sensitivity analysis (GSA) was conducted in order to explore the influence of variations in input parameters on model outputs [55] and to identify the parameters that have a significant impact on wound closure analysis. The GSA workflow proposed by Wang et al. [55] was used and structured in three phases: pre-analysis (for preparing the basic input sampling dataset), analysis (for performing sensitivity analysis and quantifying the distribution of the sensitivity index), and post-analysis (for producing the final summarized parameter ranking).

To generate random sets of the parameter value, model inputs were defined according to the Latin Hypercube Sampling (LHS) method [56], for which model simulations were performed to calculate model outputs. Specifically, the LHS method was implemented to generate 500 random sets of input parameters, and model simulations were performed to obtain the corresponding 500 sets of model outputs. To evaluate the relative effect of model parameters, a linear relation was assumed between the input parameters and output variables. The regression coefficients, $\beta_{I}$, were estimated using multiple linear regression analysis (MLRA) according to the expression equation $\mathrm{Output}=\beta_{0}+\beta_{\rho }\rho +\beta_{T_{m}}T_{m}+\beta_{T_{d}}T_{d}$. To obtain a distribution of regression coefficients, the procedure was repeated 10 times with 10 independent sets of 500 simulations (*p*-value < 0.05; data were reported in the Appendix A).

Since the dimensions of both the wound length, b_0_, and the cells, d, were only geometric parameters for our simulation, they were excluded from this analysis and fixed to the mean value (575 μm and 20 μm, respectively) obtained from the values reported in Table 1. The distribution of regression coefficients $\beta_{I}$ is reported for α, $T_{\mathrm{closure}}$, and $T_{\mathrm{half}}$ in Figure 2a–c, respectively, related to each input parameter. Through the application of statistical analysis, with one-way ANOVA and Tukey’s test (confident interval 95%), parameters were ranked according to their significance to the output (from 1 (high significance) up to 3 (low significance)), as shown in Figure 2d.

This work revealed that T_m_ was ranked as the most significant parameter for all the outputs, implying a stronger dependency of the outputs on it. No significant differences between the effects of ρ and T_d_ on the rate of closure a were revealed. On the contrary, ρ was more significant (rank 2) to T_half_ than T_d_ (rank 3), suggesting the effect of the density being higher in the earlier steps of the process, when the domain was poorly occupied by cells, while at later stages, the influence of density was limited. An opposite trend was observed for the effect of T_d_, which was more significant at later stages (T_closure_) with respect to the initial steps (T_half_) of the process.

Thus, going through the variation in the biological values of T_m_ and T_d_ (Table 1), assuming a cell type has the lowest doubling time and the highest migration time, the probability to proliferate each time step was only 4% (see Appendix A). The idea that, in the case of concurring mechanisms, the fastest one (governed by the shortest characteristic time) was also controlling the rate of the entire process was a clear concept from the analysis of the process based on the transport phenomena approach [57]. An electric analogy simplification would consider the two concurring mechanisms as two resistances acting in parallel and driven by the same driving force, which in this case was the difference between cell density at confluence (or at least in the bulk of the tissue, far from the wound) and the density in the wound, which in this case was 0. A wider analysis of the role of transport phenomena in the process was reported in a previous paper [5].

## 3. Results

In this section, we present the results of the numerical model simulations, which aimed to study the evolution of wound healing.

### 3.1. Baseline Model Behavior and Model Calibration

A direct comparison between in silico (a) and in vitro (b) experiments (along the columns) at two different times (0 and 9 h, along the rows) was reported in Figure 3. In detail, a representative phase-contrast microscopy image showing the HT-1080 wound closure process was reported and compared to a snapshot of the CA experiments. In both cases, the first time step (t = 0 h) showed a cell-free central domain populated at later times by cells migrating and proliferating up to wound closure (t~9 h). To quantify the agreement, the wound evolution of the in vitro (circle symbols) and in silico (solid line) experiments was compared in terms of A/A_0_, as reported in Figure 3c. A strong agreement in terms of α was clearly observed.

As shown in Figure 4, to assess the generality of our model, quantitative comparisons between the in vitro and in silico experiments were repeated for four different cell lines (HT-1080 human fibrosarcoma cells, MDA-MB-231 and MDA-MB-468 breast cancer cell lines, and HaCaT human keratinocyte cell line) at different cell densities. For brevity, only four conditions were reported; all input parameter values are reported in Table 2.

To confirm the goodness of the agreement, Pearson correlation coefficients were calculated (average value ~0.98), proving the reliability of our CA (see Appendix A). As a result, our model demonstrates the ability to predict the dynamic evolution of living tissues. Consequently, it can be utilized to investigate processes related to cell migration and proliferation, including cancer invasion. This model represents a promising and potent tool for identifying strategies to treat cancers more effectively.

### 3.2. Model Predictions

The CA model presented here has the capability to accurately predict migration rates and can serve as a valuable tool for investigating the dynamic evolution of complex biological systems, effectively creating a Digital Twin of the process under examination. In this section, the model was leveraged to predict a wide range of potential experimental conditions.

In detail, as shown in Figure 5, α was plotted as a function of the non-dimensional parameter $\phi =T_{m}/T_{d}$. The sets of simulations run varying values of input parameters according to the range reported in Figure 5a, and differing Thiele module Φ spanned a range [10^−4^–0.5], defined by physiological limits. As regards the minimum physiological value of $T_{d}$, it is related to the time necessary for the duplication of DNA (S1 phase), which in eukaryotic cells takes typically about 10–12 h [62]. On the other hand, $T_{m}$ can typically vary from a few seconds to a few hours in the case of poorly motile cells, such as osteoblasts [59].

As shown in Figure 5b, values of α estimated by CA were reported with respect to Φ for a set of 2000 simulations (convergence in the number of simulations was reported in the Appendix A). The data trend can be phenomenologically described by a power law (solid line) whose parameters a and b were obtained by data fitting (details reported in the Appendix A) and estimated to be $2\cdot 10^{-3}$ and 0.66, respectively:(1)$$\alpha =a\;{\left(\Phi \right)}^{-b}$$

As shown, Equation (1) is in good agreement with the experimental data, qualitatively predicting how proliferation and migration phenomena affect the closure velocity. Indeed, Φ, being defined as T_m_/T_d_, compares the contributions of migration and proliferation processes to wound healing. To elaborate further, a low Φ value, along with a decreased T_m_, indicates that cells require less time to migrate from one site to another. This signifies that these cells display high migratory behavior and, consequently, a greater potential for invasiveness.

It is worth noting a minor discrepancy observed in the high Φ range (Φ > 0.1). Specifically, the predictions from the fitting curve were consistently lower (approximately 0.25 times) compared to the simulated data points. It is important to highlight that most of the physiological conditions documented in the literature (as also seen in Table 1) predominantly corresponded to lower values of Φ, with only rare exceptions involving extremely poorly motile cells (characterized by extremely high T_m_, such as osteoblasts) [59,60]. Consequently, this high Φ region might be of marginal interest in practical applications.

To compare the prediction curve with the in vitro data, in Figure 5c, simulated and experimental values of α were reported as a function of Φ. In detail, the green circles are derived from data previously published [5,31,49,50,51,52] by our group, while the orange circles are based on other works from the literature [18,58,59,61,63,64]. The pink circles are related to a new unpublished experimental campaign, here presented for the first time.

Our prediction curve was a simple and reliable tool to quantify the migration rate of cells. The main result of this analysis, obtained from an application of our CA as a DT of the WH assay, was that, in almost any physiologically relevant condition, the WH process was driven by cell motility, while cell proliferation played a role only in rare conditions that were possible to predict by a preliminary estimation of the value of parameter Φ. For any condition corresponding to a Φ lower than a critical value of 10^−2^, cell proliferation can be neglected. The knowledge of this information, here clearly quantified for the first time, to the best of our knowledge, can simplify most of the experimental protocols.

## 4. Conclusions

Metastasis, which is the leading cause of death among cancer patients, is primarily attributed to the migration of tumor cells—a process that remains not fully understood. In silico models offer cost-effective and time-efficient tools for investigating the dynamic evolution of cell tissues in complex phenomena like metastasis. These models take into account the interplay of various biological processes, such as cell proliferation, cell motility, and cell–cell interactions.

In this work, a CA model developed in MATLAB was proposed to study cell migration by simulating the dynamical evolution of wound healing *in vitro*, and it was used as a Digital Twin of WH phenomena. The CA model was calibrated and validated by direct comparison with experimental data available in the literature, proving to be in excellent agreement with the experiments.

The main output of the application of this model to the WH was a simple power-law function able to predict wound closure rate, which was also in excellent agreement with previous experimental measurements. The proposed power-law function allowed for the identification of a critical value of a simple non-dimensional number, calculated from physiological parameters, and was able to discriminate if the wound closure process could be considered to be driven only by cell motility, neglecting the contribution of cell proliferation. This information, easy to estimate preliminarily, can allow a simplification of experimental protocols with reduced costs and times.

The methodology here proposed, based on the CA here presented, can be extended to the investigation of other biological processes. The model was easily scalable to include other phenomena, such as contact guidance or the interaction of cells with the extracellular matrix (with a different stiffness) and the presence of chemical and mechanical stimuli. The CA model is proposed and validated in a 2D application. It is worth mentioning that 2D cell culture models on solid substrates may not always adequately represent the complexity of in vivo conditions, given their limitations in replicating essential factors found in vivo, such as 3D architecture, stromal cells, and extracellular matrix [65,66,67,68]. However, a relevant part of the experimental investigation in cancer research is still performed using 2D models, at least in the initial stages of the research, with WH being a strongly popular and diffused assay. For this reason, we believe our CA model can be of great value to support the analysis and interpretation of experimental investigations. Furthermore, the proposed methodology can be extended to 3D models like spheroids or biopsies.

## Figures and Tables

**Figure 1 cancers-15-05284-f001:**
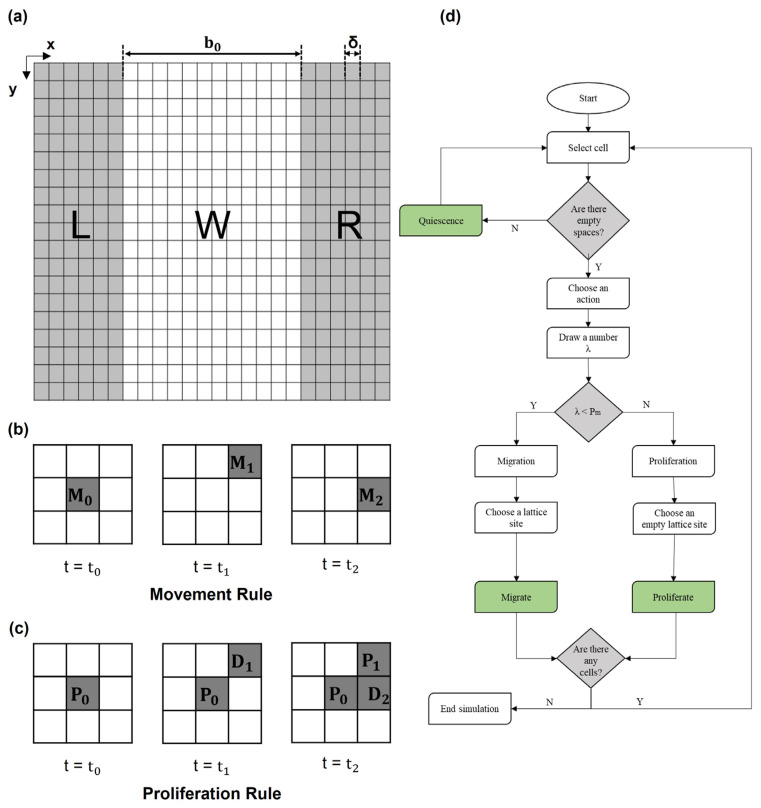
(**a**) Initial configuration of domain. At t = t_0_, the lattice domain was divided into two lateral domains (L and R, left and right, respectively) occupied by cells (each cell occupies a squared element of size δ) representing the edge of the wound and a central cell-free domain (W, wound) of size b_0_ along the x-direction. (**b**) Scheme of migration rule. A total of 3 × 3 lattice squares where automaton rules were applied: cell in the center of the square ($M_{0}$) decides to migrate; in the following steps (t_1_ and t_2_), it can move to one of the empty adjacent spaces, such as $M_{1}$ or $M_{2}$. (**c**) Scheme of proliferation rule. Cell identified with $P_{0}$ decides to proliferate; in the following step (t = t_1_), it proliferates, and the daughter cell (referred to as $D_{1}$) occupies a space in the neighborhood. At a later time, t = t_2_, the daughter cell may proliferate again, and its daughter cell, referred to as $D_{2}$, occupies another space. (**d**) Flow chart of the model algorithm.

**Figure 2 cancers-15-05284-f002:**
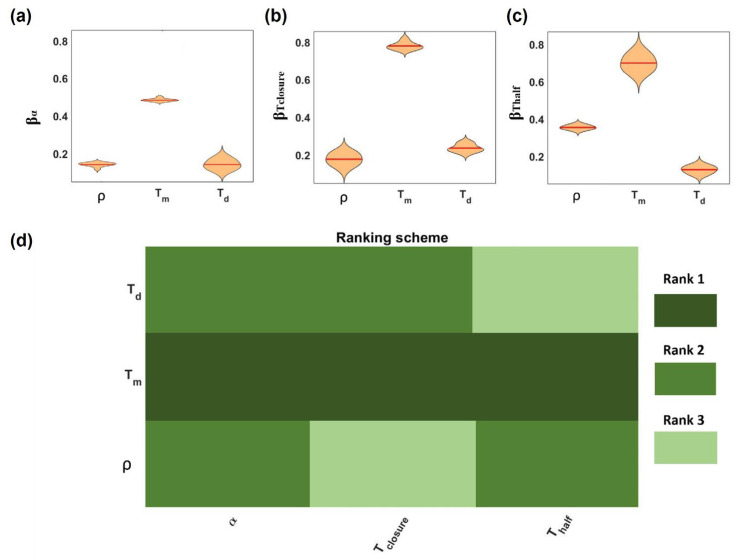
Global sensitivity analysis results. The violin plots show the regression parameter distribution (mean value reported as red line) obtained from MLRA with respect to input parameters (**a**) α, (**b**) T_closure_, and (**c**) T_half_. The weighted parameter ranking (**d**) was obtained from the weighted average of rankings from ANOVA and Tukey’s test. Note: Rank 1 (dark green) denotes the parameter of the highest significance and rank 3 (light green) the parameter of the lowest significance.

**Figure 3 cancers-15-05284-f003:**
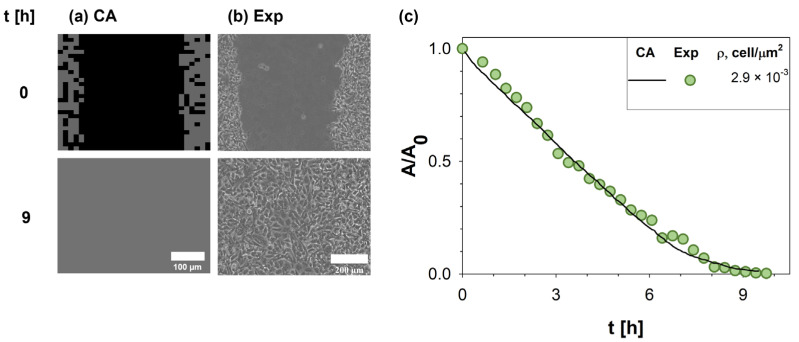
Comparison of results obtained by our CA with experimental measurements. On the left (**a**), a snapshot of in silico experiments was reported and compared to in vitro images acquired with Time-lapse microscopy during WH assay of HT-1080 at two different times (0 and 9 h) (**b**). Simulations were run using input parameters estimated from the experimental setup (Id 2 in Table 2). The wound area variation A normalized to the initial wound area A_0_ was monitored in time (**c**), providing a direct comparison between experimental data [5] (green circle symbols) and CA outputs (purple solid line).

**Figure 4 cancers-15-05284-f004:**
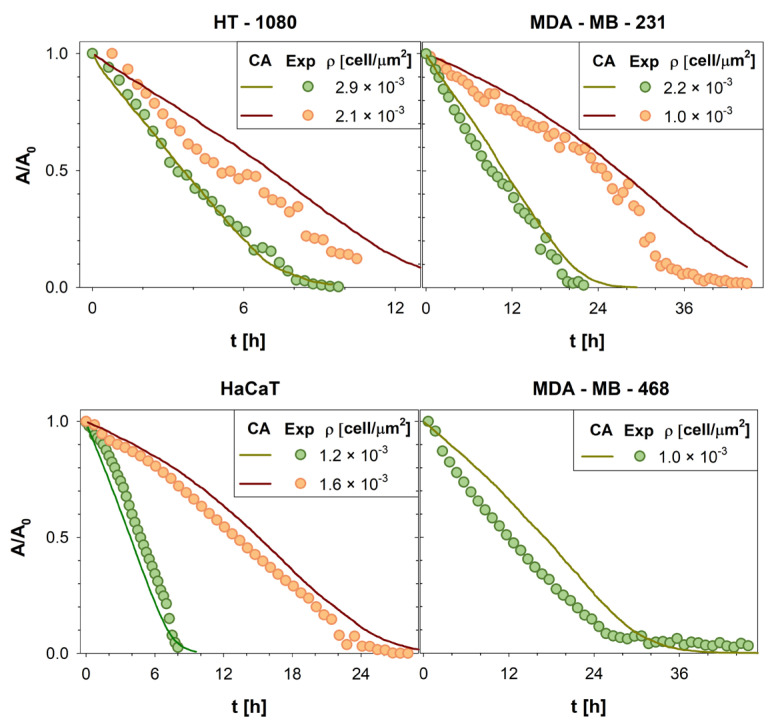
Wound area A, normalized with respect to the initial wound area A_0_, was reported as a function of time t. The evolution of the wound area in silico (green and red solid lines) was compared with the wound area variation in vitro (green and orange circles symbols) computed for all cell lines here investigated. For brevity, only four cell lines (HT1080 [5], MDA-MB231 [31,51], HaCaT [51], and MDA-MB468 [49,50]) and 7 densities were reported.

**Figure 5 cancers-15-05284-f005:**
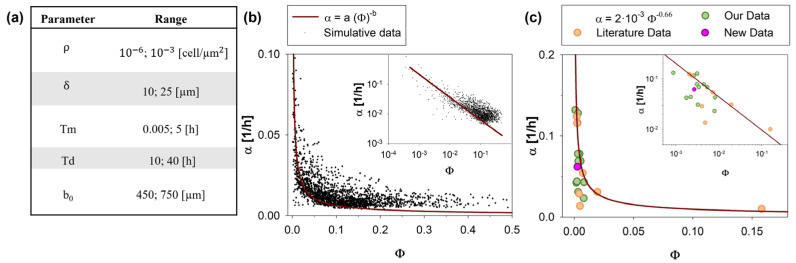
Wound area reduction rate, α, as a function of the ratio between the characteristic time of migration and proliferation, Φ. A total of 2000 in silico experiments with input parameters randomly chosen in the ranges reported in (**a**) provided outputs (black dots in (**b**)) that show power-law dependency of α with respect to Φ. The fitting curve was reported as a solid line. The inset reports the same data on a log scale. In (**c**), the trends estimated by our analysis (solid line) were compared with experimental data (our data: green circles; literature data: orange circles; new data: pink circles), reported in Table 2.

**Table 1 cancers-15-05284-t001:** List of parameters relevant to the model. ρ, δ, T_m_, T_d_, and b_0_ were input parameters, while the model allows calculating D, v, and α, which have been used to compare in silico and in vitro data. The range of variation in ρ was chosen to define the degree of coverage of the domain. The minimum corresponds to a single cell in each sub-domain (L or R), and the maximum value was associated with full coverage of the L and R sub-domains. The values of other parameters depend on the specificity of the cell line. Ranges of variation have been defined from information available in the literature [5,31,49,50,51,52] to validate the model. The details of the calculations performed to estimate parameters were reported in the Appendix A.

Parameter	Description	Range of Variation	Dimension
$T_{m}$	Characteristic time of migration	0.005; 0.5	h
$T_{d}$	Characteristic time of proliferation	12; 40	h
ρ	Density: number of cells in unit area	10^−6^; 10^−3^	cells/µm^2^
δ	Characteristic dimension of the cell	15; 25	µm
b_0_	Initial length of the wound	370; 900	µm
D	Motility: the time necessary to travel a length equal to delta	10^3^; 10^4^	µm^2^/h
v	Velocity of the fronts of cells	5; 60	µm/h
α	Velocity of wound area variation	0.02; 0.13	1/h

**Table 2 cancers-15-05284-t002:** Characteristic parameters of cell lines used for validation of CA. Ids 1–20 were derived from previous works by some of the co-authors of this paper or were taken from the literature. Id 21 presents new data from a new experiment reported here for the first time. Details about parameter estimates (*) derived from raw data available in the original papers were reported in the Appendix A. N/A= Not available; the values of some parameters were not reported due to a lack of specific information in the original papers.

Cell line	Id	ρ [#cells/μm^2^]	α [1/h]	b_0_ [μm]	T_d_ [h]	T_m_ [h]	References
**HT-1080**	1	2.7 × 10^−3^	0.012	468 *	24	0.063	[5]
	2	2.9 × 10^−3^	0.128	371 *	24	0.075	
	3	1.6 × 10^−3^	0.078	532 *	24	0.107	
	4	2.1 × 10^−3^	0.078	638 *	24	0.075	
	5	1.5 × 10^−3^	0.069	548 *	24	0.129	
	6	1.2 × 10^−3^	0.069	687 *	24	0.082	
	7	N/A	0.110	288	24	0.110	[58]
**MDA-MB-231**	8	1.0 × 10^−3^	0.023	800	38	0.338	[31,51]
	9	1.2 × 10^−3^	0.042	930	38	0.075	
	10	2.3 × 10^−3^	0.044	800	38	0.095	
	11	N/A	0.040	288	38	0.476	[58]
**MDA-MB468**	12	1.2 × 10^−3^	0.031	800	47	0.154	[51]
**HaCaT**	13	1.2 × 10^−3^	0.043	900	19	0.156	[49,50]
	14	1.7 × 10^−3^	0.132	900	19	0.017	
	15	2.5 × 10^−2^ *	0.029	N/A	19	0.078	[21]
**Saos-2: HTB 85**	16	N/A	0.010	800	37	5.851	[59,60]
**Caco-2**	17	1.2 × 10^−3^ *	0.014	882 *	80	0.385	[61]
**BEAS**	18	N/A	0.054	500	26	0.188	[18]
**MCF-7**	19	N/A	0.031	500	38	0.741	
	20	N/A	0.040	287	38	0.54	[58]
**NIH/3T3**	21	1.3 × 10^−4^	0.062	933	20	0.002	

## Data Availability

The data presented in this study are available on request from the corresponding author.

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
