# Peer review of "Exploring Cell Migration Mechanisms in Cancer: From Wound Healing Assays to Cellular Automata Models"

_cancers, 2023, doi:10.3390/cancers15215284_

Round 1

Reviewer 1 Report

Comments and Suggestions for Authors

Migliaccio et al. suggest a model for predicting the 2D collective migration of cells in the context of the commonly used wound healing assay. The discrimination between the various processes contributing to wound closure in the system used is of great value and authors suggest that the model they developed can provide such information.    

Major comments

·       Please review the title as it can be misleading, since a wound healing assay is very far from representing the cancer cell migration observed in the context of a tumour.

·       In the abstract (line 29) the authors state that “a wound healing assay was used to explore cancer cell migratory and invasive behaviour in the context of metastasis”.

Contrarily to what is stated in this sentence the authors do not analyse invasion at all, since this process that does not occur in a wound healing assay.

Additionally, the cell migration analysed is not done in the “context of metastasis” since this is an in vitro assay performed in a 2D context.

Since this sentence defines the overall objective of this work, it needs to be rewritten since, as it stands, it is misleading and not correct.

·       Despite the authors stating that that the model they developed can discrimination between the various processes contributing to wound closure (namely migration and proliferation), this is not clear in the results section. So far as I could understand, their relative contribution to would closure is not quantified, what would be of greatvalue to this field.  

Minor comments:

·       Line 65: Other common strategies include FBS % reduction (or total removal) in media (to decrease proliferation rate) or the use of shorter timepoints (8-12h) for wound size measuring after wound (to reduce the impact of proliferation in would closure).

·       Line 323, 405, 314, 372: information is missing and was replaced by “Error! Reference Source not Found”

Comments on the Quality of English Language

No comments

Author Response

Reviewer 1

Migliaccio et al. suggest a model for predicting the 2D collective migration of cells in the context of the commonly used wound healing assay. The discrimination between the various processes contributing to wound closure in the system used is of great value and authors suggest that the model they developed can provide such information.   

We thank the reviewer for their analysis of our paper. We fully agree the discrimination between different processes contributing to the wound closure is a relevant issue, and the reviewer is right identifying this as the main focus of our paper.

Major comments

  1. Please review the title as it can be misleading, since a wound healing assay is very far from representing the cancer cell migration observed in the context of a tumor.

Thank you very much for your valuable comment regarding the revision of our paper's title. We have taken your suggestion into account and agree that the term "reveal cell migration in cancer" may be misleading. We have revised the title of our paper to make it more accurate and representative of its content. The new title is as follows: “Exploring Cell Migration Mechanisms in Cancer: From Wound Healing Assays to Cellular Automata Models”.

  1. In the abstract (line 29) the authors state that “a wound healing assay was used to explore cancer cell migratory and invasive behavior in the context of metastasis”. Contrarily to what is stated in this sentence, the authors do not analyze invasion at all, since this process does not occur in a wound healing assay. Additionally, the cell migration analyzed is not done in the “context of metastasis” since this is an in vitro assay performed in a 2D context. Since this sentence defines the overall objective of this work, it needs to be rewritten since, as it stands, it is misleading and not correct.

We sincerely appreciate the feedback you've provided regarding the statement in the abstract. We understand your concerns regarding the sentence in the abstract, which might not be adequately supported by our results, even if we believe our findings are relevant for a potential (future) application to cancer metastasis invasion. We revised sentence as follows: “Methods: In this study, a wound healing assay was employed to investigate cancer cell migratory behavior, with the aim of utilizing migration as a biomarker for invasiveness.”

  1. Despite the authors stating that that the model they developed can discriminate between the various processes contributing to wound closure (namely migration and proliferation), this is not clear in the results section. So far as I could understand, their relative contribution to would closure is not quantified, what would be of great value to this field.

We agree the quantification of the contribution of migration and proliferation processes to wound closure is relevant. In our work, we proved a quantification of the relative contribution of proliferation and migration can be achieved through the Thiele module, defined as Φ = Tm/Td. This is also in line with our previous experimental work (http://dx.doi.org/10.1016/j.ces.2016.11.014). In this paper we provided a clear rational of this idea, benefitting of the CA model to obtain a wider and significant investigation. We explained more explicitly this aspect in the revised manuscript, line 397: “As shown, Equation 1 was in good agreement with experimental data, qualitatively predicting how proliferation and migration phenomena affect the closure velocity. Indeed, Φ, being defined as Tm/Td, compares the contributes of migration and proliferation processes to wound healing. To elaborate further, a low Φ value, along with a decreased Tm, indicates that cells require less time to migrate from one site to another. This signifies that these cells display a high migratory behavior and, consequently, a greater potential for invasiveness.”

Minor comments

  1. Line 65: Other common strategies include FBS % reduction (or total removal) in media (to decrease proliferation rate) or the use of shorter timepoints (8-12h) for wound size measuring after wound (to reduce the impact of proliferation in would closure).

Thank you for your suggestion, this sentence has been revised and inserted into Introduction, line 71:
“In addition, other common strategies include reducing the percentage of FBS in the media, in order to decrease the proliferation rate, or employing shorter timepoints (e.g., 8-12 h) for measuring wound size after the initial wound creation. This approach minimizes the impact of cell proliferation on wound closure.”

  1. Line 323, 405, 314, 372: information is missing and was replaced by “Error! Reference Source not Found”.

We've addressed the missing references issue that you pointed out. The missing information, previously labeled as "Error! Reference Source not Found," has now been corrected.

Reviewer 2 Report

Comments and Suggestions for Authors

PFA

Author Response

Reviewer 2

Comment to the article: A Comprehensive Approach to reveal cell migration in cancer: from Wound Healing experiments to Cellular Automata models.”

The article provides a global sensitivity analysis to explore the influence of variations in input parameters on model outputs. This analysis helps identify the parameters that have a significant impact on wound closure analysis, providing valuable insights for further research. However, there are some opened questions which still need to be answered prior to the application of this study for cancer treatment.

  1. The study comprises of four different cell lines, which may not fully represent the diversity of cancer types and their behaviors. This limited sample may affect the generalizability of the findings. How do the authors provide a rationale for these limitations.

Thank you for pointing this out.

Our study encompasses nine different cell lines. We were probably not clear enough claiming this aspect. Specifically, we included four distinct cell lines from our research group (HT-108, MDA-MB-231, MDA-MB-468, and HaCaT) to calibrate our Cellular Automata model with regards to wound closure. Subsequently, the model was validated using five additional cell lines, four of which were sourced from existing literature (Saos – 2: HTB 85, Caco – 2, BEAS and MCF – 7), and one from a novel, unpublished experimental campaign (NIH/3T3). These cell lines were deliberately selected to ensure a extended validation across a diverse range of cell types and tissues. In fact, the validation encompassed epithelial, and fibroblast cells, originating from both murine and human sources, and encompassed both tumoral and non-tumoral cells. More information has been added to the introduction.

  1. The model was validated using in vitro experimental data, which may not fully capture the complexity of in vivo The differences between in vitro and in vivo environments could introduce uncertainties in the model's predictions.

Thank you for raising this important point. We acknowledge that 2D cell culture models on solid substrates may not fully replicate the complexity of in vivo conditions, given their limitations in replicating essential factors found in vivo, such as 3D architecture, stromal cells, and extracellular matrix.

Our methodology primarily focuses on investigating and predicting cell migration and invasiveness. While the results reported in this paper were focused on 2D models, the methodology presented can be further developed and applied also to 3D models, offering a more comprehensive representation of the in vivo environment.

We agree with the reviewer about the higher value of 3D models in better capturing the complexity of in vivo tumors. However, we can state that still nowadays a relevant part of the experimental investigation in cancer research is still done using 2D models, at least in the initial stages of the research, WH being a strongly popular and diffused assay. For this reason, we believe a tool, like the one we are proposing, even at the stage of development we are here presenting, can be of value to support the analysis and interpretation of experimental investigations.

This observation has been added to the conclusions (line 469).

  1. The range of input parameters used in the simulations was defined by physiological limits. However, this range may not encompass the full spectrum of possible parameter values, potentially limiting the model's ability to capture the full range of biological behaviors.

Thank you for bringing up this point. The range of input parameters investigated in this work was estimated taking into account physiological limits available in the literature, as reported in Table 1. This reflects the values that have been commonly used and observed in experiments reported in the literature. These values provide a foundation for our simulations and help us align our model with the observed biological behaviors as documented in the existing research. However, we can add that the investigation we run in the GSA and LSA was extended even above the minimum and maximum registered values of each of the parameters, still limiting the investigation to the orders of magnitude of reasonable validity for cell tissues. We are not reporting here data that could not be validated by a sound comparison with experimental results, but the model can be potentially used also to investigate other ranges of values, even outside the limits now reported in the literature.

  1. The model assumes that the fastest mechanism, governed by the shortest characteristic time, controls the rate of the entire process. While this assumption simplifies the analysis, it may not fully reflect the complex interplay of multiple mechanisms involved in cancer invasion and wound healing.

Thank you for highlighting the assumption we made. We acknowledge that this assumption simplifies the analysis and may not fully capture the complex interplay of multiple mechanisms involved in cancer invasion, in particular when in vivo applications are of interest, as widely discussed above. However, it is worth noting that despite this simplifying assumption, our model demonstrated good agreement with experimental data. We compared experiments at different cell densities, and with 9 different cell lines, with the aim to mimic different conditions, in the entire range of parameters investigated the model's predictions aligned well with the observed outcomes.

  1. Reference no. 51 and 53 are same it should be corrected.

Thank you for pointing out the duplication in references 51 and 53. We have addressed this issue, and the references have been corrected accordingly.

Round 2

Reviewer 1 Report

Comments and Suggestions for Authors

The authors have addressed all issues raised in my first review report.